# Auriferous Quartz Veining Due to CO$_2$ Content Variations and Decompressional Cooling, Revealed by Quartz Solubility, SEM-CL and Fluid Inclusion Analyses (The Linglong Goldfield, Jiaodong)

**Qing Wei [1,2], Hongrui Fan [3,4,\*], Jacques Pironon [5] and Xuan Liu [5,\*]**

[1]   Institute of Crustal Dynamics, China Earthquake Administration, Beijing 100085, China;
      weiqing@email.eq-icd.cn
[2]   State Key Laboratory of Ore Deposit Geochemistry, Institute of Geochemistry, Chinese Academy of Sciences,
      Guiyang 550081, China
[3]   Key Laboratory of Mineral Resources, Institute of Geology and Geophysics, Chinese Academy of Sciences,
      Beijing 100029, China
[4]   College of Earth Science, University of Chinese Academy of Sciences, Beijing 100049, China
[5]   Université de Lorraine, CNRS, CREGU, GeoRessources, F-54000 Nancy, France;
      jacques.pironon@univ-lorraine.fr
\*   Correspondence: fanhr@mail.iggcas.ac.cn (H.F.); xuan.liu@univ-lorraine.fr (X.L.);
      Tel.: +86-010-8299-8218 (H.F.); +33-037-274-5575 (X.L.)

**Abstract:** Quartz is the most common gangue mineral in hydrothermal veins. Coupled with capacities of hosting fluid inclusions and recording varieties of microtextures, its solubility behavior may provide unparalleled insights into hydrothermal processes. In this study, the Linglong goldfield in Jiaodong is targeted to investigate gold-producing quartz veining process. Scanning electron microscope (SEM)-cathodoluminescence (CL) imaging uncovered three episodes of quartz deposition, intervened by an episode of quartz dissolution. Based on newly-developed quartz solubility diagrams and CL-aided fluid inclusion microthermometry, it is proposed that precipitation of the earliest quartz (Qz1) was controlled by CO$_2$ content increase and subordinately affected by decompressional cooling, leading to the formation of the early thick gold-barren veins (V1); the second generation of quartz (Qz2a) was formed by the same fluids that may have been diluted and cooled by meteoric water, leading to a greatly reduced quantity of quartz and the deposition of pyrite and gold; and the third generation of quartz (Qz2b) was deposited along with polymetallic sulfides, due to fluid cooling following a quartz dissolution event likely induced by cooling in retrograde solubility region and/or CO$_2$ content decrease. This research may elucidate gold formation processes in orogenic intrusion—related deposits, and points to imperative CL-based in situ analyses for future studies.

**Keywords:** mineral solubility; hydrothermal veins; Jiaodong gold; fluid overprint

## 1. Introduction

Quartz is the most important silica mineral and widely occurs in igneous, metamorphic, sedimentary and hydrothermal rocks of the Earth's crust. Because of its abundance and properties, quartz finds many applications in industry and scientific researches [1]. For instance, its luminescence, geochemical properties and hosting fluid/mineral inclusions have provided critical knowledge for reconstructing geological processes [2–5]. In hydrothermal ore deposits, quartz is the most common gangue mineral and may intergrow with metals (such as native gold) and sulfides containing metals such as copper, tin, and tungsten [6]. The veining processes of quartz and associated minerals in

porphyry Cu deposits have been intensively studied and well understood with the analytical aid of cathodoluminescence (CL) petrography [7–9], solubility analysis [10,11], fluid inclusion [12,13], trace elements [14] and oxygen isotopic composition [15,16]. For instance, highly complex veining processes involving the deposition, dissolution, reopening, recrystallization and overgrowth of quartz and incursion of cooler and dilute meteoric waters have been discovered [8,15]. These pieces of information ultimately shed lights on where cupriferous fluids are sourced and how these fluids precipitate metal sulfides and quartz. By contrast, quartz solubility analysis has been much less employed in the studies of orogenic intrusion-related gold deposits, leading to less knowledge of the veining processes that are responsible for gold deposition. Several previous studies were based on quartz solubility in pure water, rather than the $H_2O$-NaCl-$CO_2$ ternary system that is a better approximation of orogenic gold fluid [17,18]. Recently, Monecke et al. [18] constructed solubility diagrams in the $H_2O$-NaCl-$CO_2$ ternary system, but data are only available for the single-phase region. This lack of data for two-phase field poses a challenge for thoroughly understanding vein formation processes in orogenic intrusion-related setting, since fluid immiscibility is quite common and is considered a critical factor for gold genesis [19].

The Linglong gold deposit located in the Jiaodong peninsula, eastern China, is targeted in this study, to fill this void of knowledge. This deposit is selected because (1) it is a vein-type gold deposit, comprising large quartz veins that host gold occurrence; and (2) its geology, geochemistry and isotopic characteristics have been well documented [20–27]. Based on a methodological combination of CL petrography, fluid inclusion and newly-developed quartz solubility diagrams that cover both the single-phase and the two-phase field, this study elucidates veining processes in this deposit, which has implications for understanding quartz deposition and dissolution processes in a wide range of geological settings.

## 2. The Linglong Gold Deposit

### 2.1. General Geology

The Jiaodong gold province (JGP) has been the subject of many detailed studies for many decades [28–31], and its geology and metallogeny have been synthesized and updated by many past and recent researchers [32–44]. Therefore, only a brief introduction closely relevant to this study is provided. The region of the JGP is a peninsula located in northeastern China (Figure 1a inset). Tectonically, it straddles the southeastern part of the Precambrian North China Craton (NCC) in the north and the Triassic Sulu ultra—high pressure orogenic belt (UHP) in the south; to the west, it is bounded by the north-northeast trending Tan-Lu fault zone [45] (Figure 1a). This region is considered to have been part of the Pacific tectonic regime since the early Mesozoic period [46,47]. The rocks exposed include basement rocks that were metamorphosed under high amphibolite to granulite facies conditions in the early Proterozoic period (1.80–1.95 Ga [48]), Triassic UHP rocks [49] and voluminous Cretaceous volcanic-sedimentary rocks [50], Triassic to Cretaceous granitoids [51] and mafic to intermediate dykes [52] (Figure 1a).

Gold deposits in Jiaodong are structurally controlled by a set of NE- to NNE-trending fault systems [53], forming three clusters, from west to east, namely the Zhaoyuan–Laizhou, Penglai–Qixia and Muping–Rushan belts [54]. Over 95% of the gold resources are hosted in the Jurassic to Cretaceous granitoids [55], with only a few small-sized deposits being hosted in the metamorphic basement [56,57]. Gold occurrence occurs either as auriferous quartz veins (Linglong-type), or as disseminated/stockwork altered rocks (Jiaojia-type) [29] (Figure 1a). The difference in mineralization styles, for instance in the Linglong goldfield, may have been caused by difference in local stress field and the related physical and chemical variations [26].

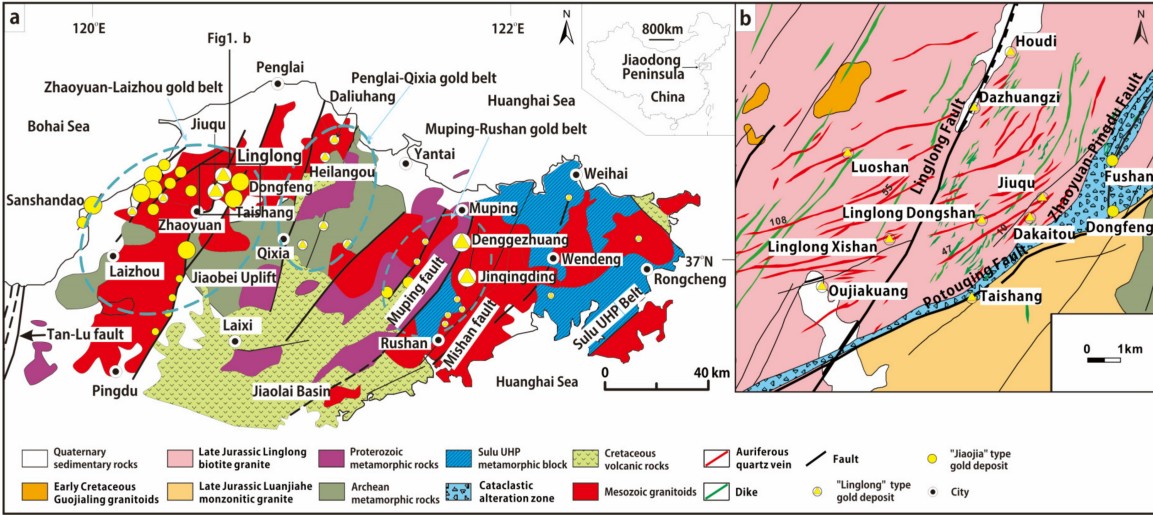

**Figure 1.** (**a**) A simplified geological map of the Jiaodong gold province, which is located in the eastern China (peninsula) (inset), with spatial distribution of the gold deposits. (**b**) A geological map of the Linglong goldfield with locations of known mining camps. Filled circles represent disseminated ore deposits; circles filled with a triangle represent vein-type deposits. Symbol sizes are proportional to gold reserves.

The Linglong goldfield, the largest gold producing area in Jiaodong, is located at the northern tip of the Zhaoyuan–Pingdu sub-belt (ZPB), which commonly hosts quartz vein type gold ores (Figure 1b). This goldfield is characterized by the presence of hundreds of gold-producing quartz veins, occurring in steeply dipping, NE–NNE trending extensional cracks in the hanging wall of ZPB. These veins are currently under exploration and production by several mines such as Dongshan, Xishan, Dakaitou, and Jiuqu. In the eastern part, the ZPB is joint by the Potouqing fault, the conjunction of which hosts disseminated ores [26]. Most of the gold deposits are usually hosted in the Jurassic Linglong biotite granite, with deep ores, for instance at Jiuqu, being hosted in both the Linglong biotite granite and Cretaceous Guojialing granodiorite [26]. The Linglong fault crosscuts the ZPB and its subordinate faults controls the emplacement of many intermediate to mafic dykes.

Over two hundred veins are exposed to the surface in the Dongshan and Xishan mines, among which roughly 30 veins have economic values, such as the veins #108, #55, #53, #52, #51, #48, #47, #10 and #9 (Figure 1b). In general, the veins mostly strike NE between 35° to 70° and dipping NW, whereas they may change to SE-dipping at shallows and NW-dipping at depth near the Potouqing fault. These veins often reach thousands of meters in length and have variable widths of one to tens of meters. Ore grades vary from a few to tens of grams per ton up to hundreds of grams per ton, where visible gold can be observed.

## 2.2. Alterations, Veins and Mineralization

Hydrothermal alteration, veins and mineralization characteristics of disseminated ore bodies have been described in detail by Wen et al. [26] and Guo et al. [27]. This section primarily focuses on illustrating vein mineralogies, alteration petrography and spatial zonation.

Hydrothermal alteration haloes observed at the Linglong goldfield include potassic (mainly biotitization and K-feldsparization), sericitic, silicic, pyritic alterations and carbonatization. Spatially, these alteration haloes commonly juxtapose to form a lateral zonation around the thick central quartz veins (Figure 2a). Potassic alterations developed distal to the quartz vein as the outer part of the alteration zones in the Linglong granite, and is often overprinted and transits gradually into sericitic, and pyritic–sericitic alterations. The alteration haloes and spatial distribution are comparable to those of other disseminated gold deposits, such as the Sanshandao deposit in the western part of the province [58] and the Dongfeng deposit in this region [26], but differ from them by having much

smaller thickness. Biotitization is represented by a significant increase in biotite volumetric contents in comparison to fresh Linglong biotite granite (Figure 2b). Even though its spatial relationship with quartz veins is still unclear, it occurs in a more widespread manner in the gold district than the surrounding areas. Biotitization might have formed slightly earlier than K-feldspar alteration (Figure 2c), which are both crosscut and superimposed by sericitic alteration (Figure 2d).

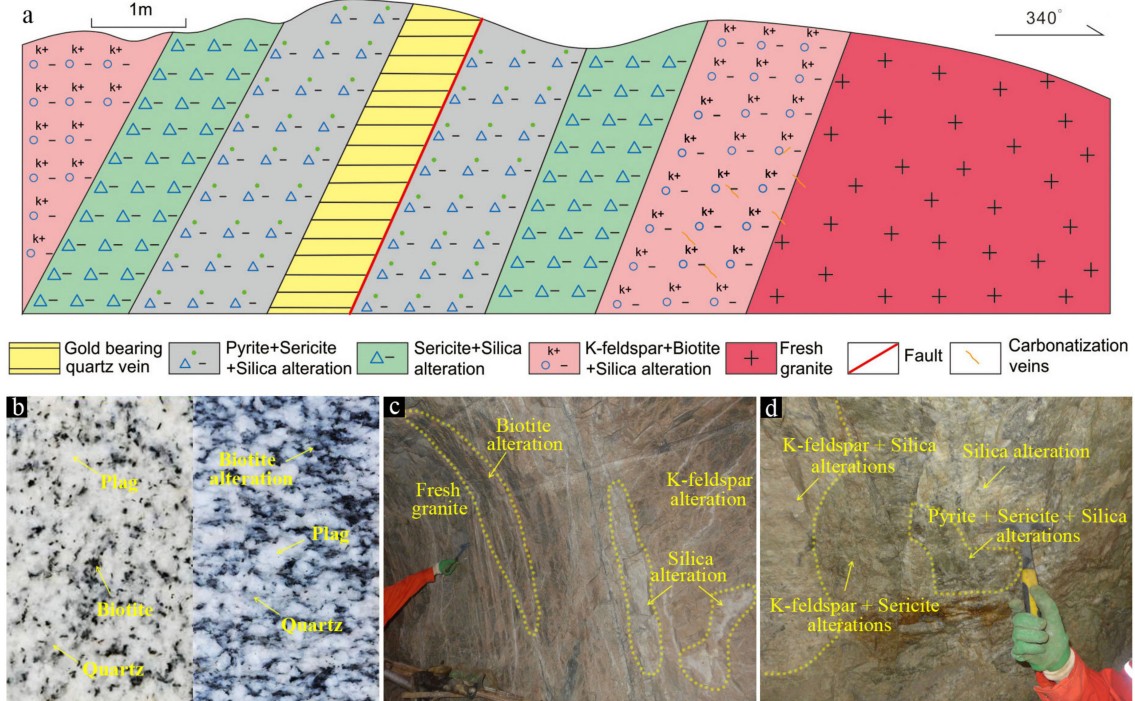

**Figure 2.** A sketch of alteration zonation (**a**) and alteration styles at the Linglong goldfiled (**b–d**). (**b**) fresh biotite granite (left) and granite with biotitization. (**c**) Pink to flesh pink potassic alterations developed in the biotite granite and were overprinted by silicic alteration. (**d**) Silicification with local pyritic alteration. Plag: plagioclase.

Based on field (underground tunnels and outcrops) and hand specimen observations and crosscutting relations, four generations of fresh veins have been identified at Linglong. From early to late, they are: quartz ± pyrite vein (V1), gold + quartz + pyrite vein (V2), gold + quartz + base mental sulfides vein (V3) and quartz + calcite ± pyrite vein (V4). V1 veins are often fractured and reopened by later veins and have potassic haloes (Figure 3b); V2 veins are usually several centimeters thick and take on liner distributions, with pyrite ± quartz ± sericite alteration selvages (Figure 3c); the cutting relationship showed that V1 were cut by V2 (Figure 3d).

In V1, quartz accounts for >95 vol% of vein minerals, with minor amounts of coarse-grained anhedral or subhedral pyrite. Little gold was precipitated with this vein type. The quartz grains are subhedral to euhedral in morphology and display milky to white color (Figure 4a). Pyrite grains are commonly fractured. V2 veins consist of about 20 vol%–30 vol% quartz and 70 vol% pyrite. Gold mainly occurs as native gold in pyrite fractures. Quartz in this vein type show white to gray color and poor transparency. Pyrite grains show euhedral cubes and semi-euhedral aggregates and often cement or crosscut V1 (Figure 4b). V3 veins are characterized by the presence of appreciable amounts of base-metal sulfides, including pyrite, chalcopyrite, galena, sphalerite, pyrrhotite and tetrahedrite. Metal sulfides occur with gold and low concentrations of quartz (≤5 vol%) (Figure 4c). Quartz gains in this type of vein are usually dark to gray in color and their relative volumetric percentage is significantly decreased in comparison to that of V1. Pyrite and other metal sulfides are fine-grained semi-euhedral and anhedral aggregates. V2 and V3 veins are the major sources of gold. V4 veins are commonly observed in granite and can also be observed in mafic dykes (Figure 4d), and pyrite and gold are rarely seen.

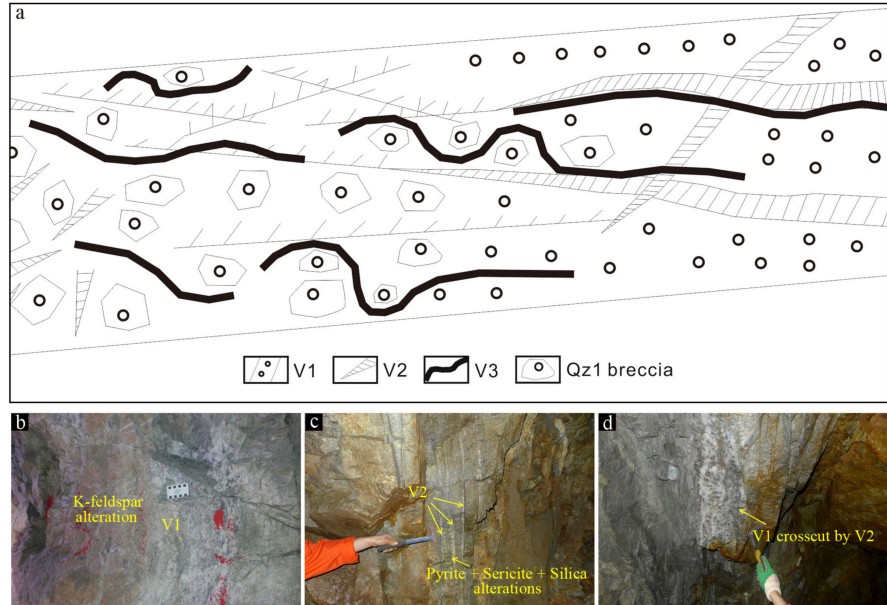

**Figure 3.** A sketch of vein types and their crosscutting relationship (**a**) and underground-tunnel images of the vein generations of the Linglong goldfield. (**b**) Cracked V1 developed with K-feldspar alterations. (**c**) V2 overprinted by the pyrite + sericite + Silica alterations. (**d**) V1 crosscut by V2.

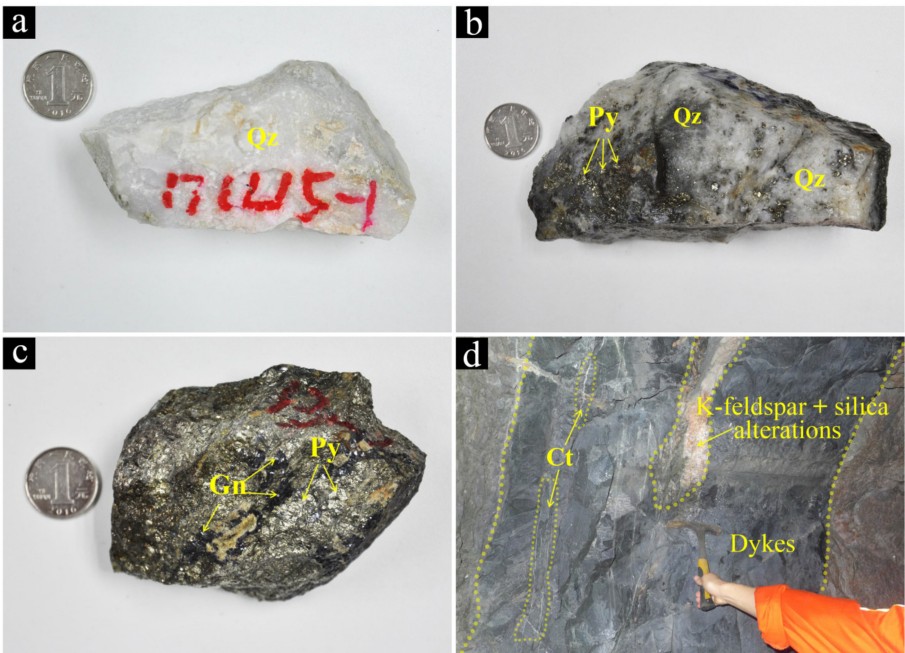

**Figure 4.** The ores and the veins in different stages of mineralization at the Linglong goldfield. (**a**) A hand specimen of V1with milky white quartz; (**b**) A hand specimen of V2 with quartz in white-gray color and poor transparency; (**c**) A hand specimen of V3 appearance of base-metal sulfides with golden color. (**d**) A tunnel picture of V4 in dykes and overprinted by potassic and silicic alterations. Qz: quartz, Py: pyrite, Ct: calcite, Gn: galena.

## 3. Analytical Techniques and Methods

### 3.1. Samples and Analytical Protocols

Twenty-nine vein and rock samples were selected from underground tunnels of the Xishan (14 samples), Jiuqu (15 samples) mines in the Linglong goldfield. Thin and thick sections were made for petrographic observations by microscopy, and eleven sections were carefully studied by a scanning electron microscope (SEM) equipped with a cathodoluminescence (CL) detector. Eight quartz samples were studied using cooling/heating experiments and laser Raman spectroscopy to determine the fluid pressure, temperature and composition. Microthermometric measurements and Raman analyses on the fluid inclusions were carried out using a Linkam THMSG 600 heating-freezing stage, combined with a Zeiss microscope and the HORIBA LabRam HR800 Raman spectrometer, respectively, at the Institute of Geology and Geophysics, Chinese Academy of Sciences (IGGCAS). The cooling/heating stage was calibrated against the triple-point of pure $CO_2$ (−56.6 °C), the freezing point of water (0 °C) and the critical point of water (374.1 °C), using fluid inclusions synthesized by FLUID Inc. Cooling/heating rates of 1 to 2 °C/min in the chamber were used until phase transition in the fluid inclusion was approached when the rates were reduced to 0.1–0.2 °C/min for more accurate temperature documentation. Carbonic phase melting ($T_{m,CO2}$) and clathrate melting ($T_{m,clath}$) were determined by temperature cycling [59]. The precision of measurements was ±0.2 °C at temperatures below 30 °C and ±2 °C at the temperature of the final homogenization. Fluid salinity and density were calculated based on $T_{m, ice}$ and $T_{m, clath}$ by using the Flincor program [60]. For the Raman analysis, an argon laser with a wavelength of 514.5 nm at power of 1000 mW were used. A range in wave number between 100 $cm^{-1}$ and 4000 $cm^{-1}$ was set for the spectrometer during analysis, which may cover peak positions for common minerals, liquids and gases.

SEM-CL imaging was undertaken at the Institute of Geochemistry, Chinese Academy of Sciences (IGCAS). The SEM was a JSM-7800F scanning electron microscope equipped with a Gatan Mono CL4 spectrometer and EDAX TEAM Apollo XL energy—dispersive spectrometer. The carbon-coated and polished samples were analyzed at 10 kV with beam current at 10 nA, and the detection range of wavelength is from 300 to 700 nm.

### 3.2. Quartz Solubility Diagrams

Diagrams of quartz solubility in this study are constructed for the $H_2O$-NaCl-$CO_2$ ternary systems. The temperature, pressure, composition ranges between 100 and 600 °C, 1 and 4000 bar, salinity of 5 wt% NaCl, and 10 mol%–15 mol% $CO_2$, respectively, based on previous [26,27,61] and present fluid inclusion studies on the Linglong goldfield.

The solubility diagrams (Figure 9, Figure 10 and Figure 13) consist of two types of information: (1) stable phase species at the P-T-X conditions under investigation; and (2) quartz solubilities at the corresponding P-T-X conditions that are plotted as isopleths (locus of equal solubility points). The phase boundaries between liquid and vapor phases are computed by using the thermodynamic model of Mao et al. [62], which allows one to the calculate $CO_2$ solubility in aqueous NaCl solutions and thus $CO_2$ saturation curve at different P-T-$X_{NaCl}$. This curve is taken as the phase boundary between liquid and vapor phases. The relatively low salinity (5 wt% NaCl) is assumed so that no NaCl saturation occurs in the selected P-T and $X_{CO_2}$ range. It should be noted that Mao et al.'s thermodynamic model is applicable to temperatures up to 450 °C and pressures up to 1500 bar. Extrapolation is made for conditions beyond these limits in this study.

Quartz solubility was computed with the thermodynamic model of Akinfiev and Diamond [63] (AK model). In their model, the most important unknown parameter is mole volume of the mixture fluids ($V_{mix}$). In this study, the $V_{mix}$ at given P, T conditions was calculated by using a predictive PVTX model of Mao et al. [64,65], which is a function of temperatures, pressures, and fluid compositions. Quartz solubilities plotted as isopleths in the diagrams (Figure 9, Figure 10 and Figure 13) were computed in two steps: first, transform the AK model into an implicit equation between *T*, *P*, *x* and

$m_{SiO2}$; second, solve each specific $m_{SiO2}$ with T-P relationships at fixed composition with Newton Iteration. Despite the presence of several other density models [66,67], AK model is selected, due to its simplicity, high accuracy and predictability [11,18].

The algorithms related to the above calculations are embedded in three Fortran programs (available at: https://www.researchgate.net/publication/341204680_Quartz_solubility_calculation_program). In these programs, for phase boundary calculation, the input parameters are temperature, mole fractions of $CO_2$, and weight percentage of NaCl; for quartz solubility calculation, the input parameters are temperature, pressure, mole fractions of $CO_2$, mole fraction of NaCl, and mole fraction of $H_2O$. For calculating quartz solubility isopleth, the input parameters are temperature, molar percentage of $CO_2$, weight percentage of NaCl, and quartz solubilities.

## 4. Results

### 4.1. CL Textures and Quartz Sequences

Quartz from the four types of vein are examined by SEM-CL imaging, and the microtextures are used as petrographic guidance for constructing mineral paragenesis and quartz sequence, and for guiding fluid inclusion petrography. These textures are presented in Figure 5 and are summarized in Table 1.

**Table 1.** Characteristics of vein types of the Linglong gold deposit.

| Vein Type | Quartz Microscopic Features | Quartz CL Features | Associated Sulfides | Associated Alterations |
|---|---|---|---|---|
| V1 | euhedral-subhedral coarse grained; milky white and purify; fragmented | bright (Qz1); dark (Qz2a) dark (Qz2b) | pyrite | potassic |
| V2 | euhedral or subhedral grains; stress; white gray and poor transparency | bright (Qz1); dark (Qz2a) | pyrite | potassic ± sericitic ± pyritic ± silicic |
| V3 | subhedral grains; dark gray and content reduced | bright (Qz1); dark (Qz2a) dark (Qz2b) | pyrite ± chalcopyrite ± galena ± sphalerite ± magnetite ± hematite ± pyrrhotite | potassic ± sericitic ± pyritic ± silicic |
| V4 | white quartz | bright, zoning | pyrite (rare) | carbonate |

Based on CL images taken from seventy-nine areas in a total of twenty vein samples, two primary textures (PT) and three secondary textures (ST) in quartz have been identified. The first primary texture (PT1) is subhedral to euhedral concentric zonation of bright and dark bands, and quartz with this feature is categorized as Qz1 (Figure 5a,k); the second primary texture (PT2) is subhedral, dark CL, and quartz with this feature is designated as Qz2. The primary texture in Qz1 can be strongly obliterated, where it is intensively fractured and dissolved by Qz2 (Figure 5a). Two subtypes for Qz2 can be distinguished according to occurrence and sulfide association. The Qz2a is represented by the quartz grains filled in relatively wider fractures (ST1), and in places, caused the brecciation of Qz1 (Figure 5c). They commonly intergrow with pyrite, gold and minor amounts of other sulfides (Figure 5b). Qz2b are those quartz gains fill in the voids networks in Qz1 (ST2) (Figure 5a,c), which are quite similar to the so-called "splatters" and "cobwebs" in early quartz grains in porphyry Cu deposits [8]. Another form of Qz2b is the pseudomorphic replacement of Qz1 (ST3, Figure 5e,g). These Qz2b quartz might have formed slightly later than the "splatters and cobweb"-filling Qz2b, but no crosscutting relation has been observed. However, a difference is that the replacement Qz2b commonly shows close spatial association with polymetallic sulfides (pyrite, sphalerite, galena, chalcopyrite, native gold and electrum), whereas the same association is absent for "splatters and cobweb"—filling Qz2b (Figure 5h,g).

Spatially, Qz1 primarily occurs in V1 quartz veins and V2 quartz-pyrite veins, but also occurs in low abundance in V3 quartz—polymetallic sulfide veins. Qz2a grains occur in variable amounts in all vein types, and are strongly associated with Qz1. Qz2a grains also occur in V1 and V2, but rarely in V3; in contrast, Qz2b grains occur primarily in V3, but rarely in V1 and V2.

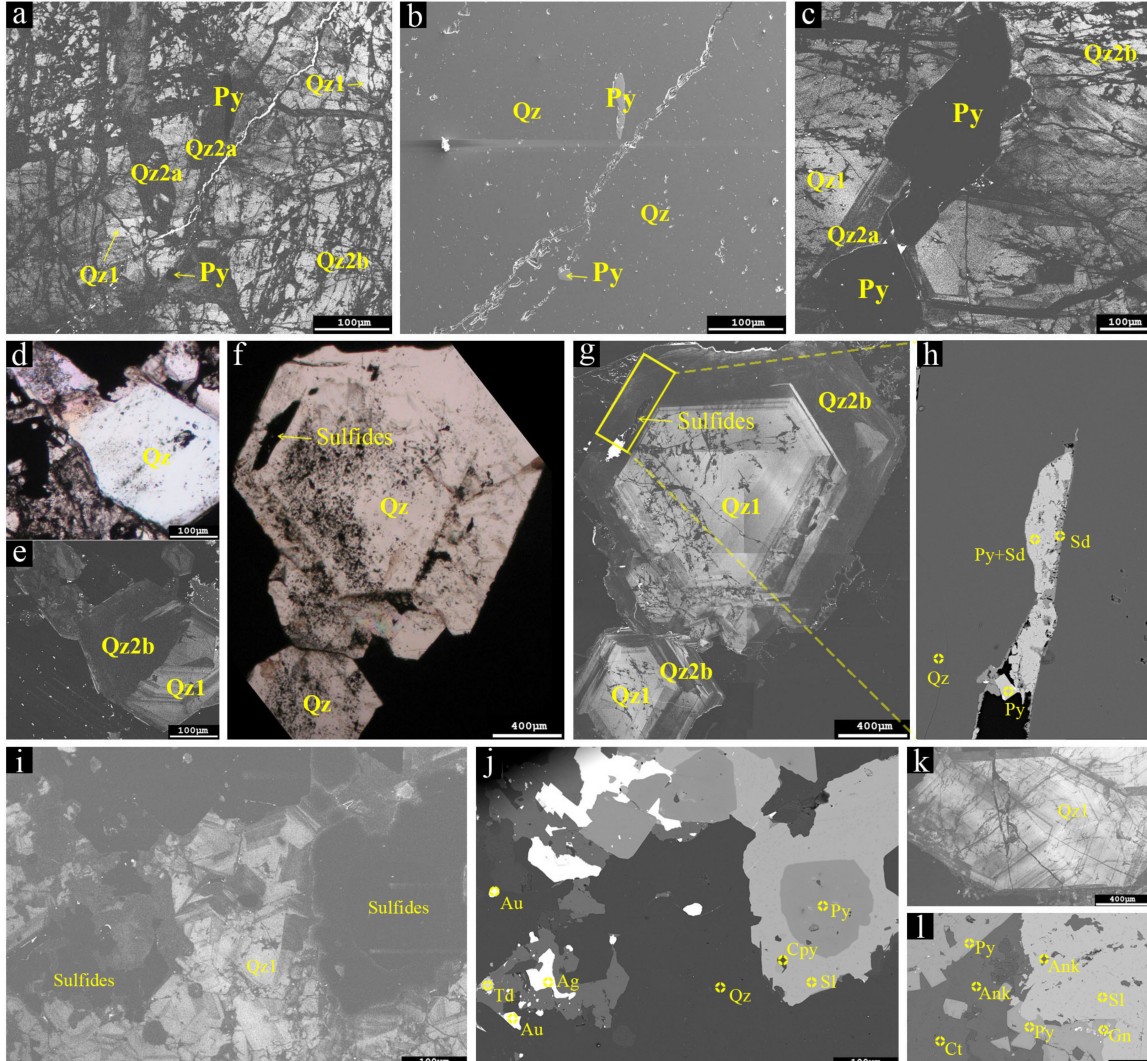

**Figure 5.** Microscopic and SEM-cathodoluminescence (CL) images of quartz samples of the Linglong goldfield. (**a**) CL-bright Qz1 crosscut by later CL-gray to dark Qz2a in V1 vein; (**b**) secondary electron image of (a); (**c**) Pyrite grain intergrown with Qz2a in a V2 vein; (**d**) Euhedral quartz under transmitted light in a V2 vein; (**e**) Qz1 recrystallized by Qz2b with the same view as (d); (**f**) Euhedral quartz under transmitted light in V3 vein; (**g**) Qz1 recrystallized by Qz2b with the same view as image (f); (**h**) Backscattered electron (BSE) image with energy dispersive spectrometry (EDS) analyses of the area enlarged from (f); (**i**) Qz2b intergrown with sulfides in a V3 vein, with abundant relict Qz1; (**j**) EDS analysis of the same area of (i); (**k**) Qz1 in aV4 vein; (**l**) EDS analysis of minerals in a V4 vein. Quartz generations are detailed in the text. Py: pyrite, Sd: siderite, Cpy: chalcopyrite, Gn: galena, Sl: sphalerite, Td: tetrahedrite, Ag: Argentum, Au: Aurum; Ct: calcite; Ank: ankerite.

## 4.2. Fluid Inclusion Petrography and Microthermometry

Based on optical properties (phase species and ratios at room conditions), four types of fluid inclusions have been distinguished in the Linglong goldfield, i.e., pure $CO_2$ (A type), $H_2O$-NaCl-$CO_2$ liquid-vapor (B type), $H_2O$-NaCl liquid-vapor (C type), and daughter mineral-bearing liquid-vapor

(D type) inclusions (Figure 6), which is consistent with previous studies [26,27]. The A-type inclusions consist of monophase $CO_2$ (vapor or liquid) (Figure 6a), and two-phase $CO_2$ ($L_{CO2}$ + $G_{CO2}$) (Figure 6b). This kind of inclusion is typically scattered in V3 and V4, with sizes of 5–10 μm. B-type inclusions commonly consist of a carbonic phase and an aqueous phase. According to the species of carbonic phase, B-type inclusions are divided into B1 ($L_{CO2}$ + $G_{CO2}$ + $L_{H2O}$) (Figure 6c) and B2 ($G_{CO2}$+$L_{H2O}$) inclusions. B1-type inclusions normally have volumetric ratio of carbonic phase 30%–40% (Figures 6c and 7a) whereas the ratios for B2-type inclusions vary between 15%–70% (Figures 6d and 7a). The B-type inclusions is widely distributed in the whole vein generations and has a relatively big size, ranging from 10–20 μm. The presence of $CO_2$ phases is confirmed by Raman spectra (Figure 8a), and no other gas species are detected, which is consistent with microthermometric measurements. C-type inclusions normally consist of a vapor phase and a liquid phase with variable V:L ratios, and are developed from V2 to V4 with an intermediate size varying from 4 to 16μm (Figure 6e). They can be subdivided into C1 (V:L > 50%) and C2 (V:L < 50% and variable) (Figure 7a). No $CO_2$ components have been detected for these inclusions by Raman spectroscopy. In D-type inclusions that only appeared in V1 and V3 veins, there exists a solid phase, which is commonly calcite, as verified by laser Raman spectra (Figure 8b), along with a vapor and liquid phase (V:L ratios of 30%). The spatial distribution of fluid inclusions in quartz is revealed by CL imaging. In Qz1, B1-type inclusions occur along growth zones and are interpreted as primary in origin, and A- and C-type inclusions are also observed on or near healed cracks (Figure 7a), which are interpreted as secondary in origin. Due to a scarcity of inclusion in the studied Qz2a, no primary fluid inclusions have been distinguished. In this case, fluid inclusions in the V2 veins were measured by microthermometry, but cautions were taken when interpreting these data. In Qz2b, typical boiling assemblages were observed, with end-members of A-type and C1-type and mixture of B2-type inclusions (Figure 7). Abundant C2-type inclusions are also present, but maybe represent later fluids, as supported by their low homogenization temperatures (Figure 7b).

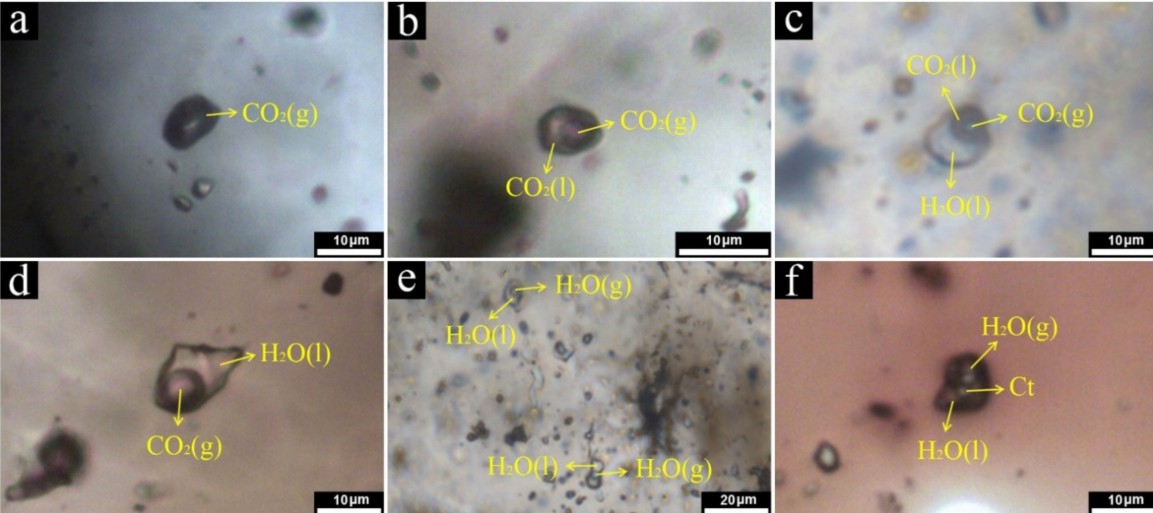

**Figure 6.** Microphotographs of different types of fluid inclusion at the Linglong gold deposit. A-type fluid inclusions with monophase $CO_2$ ($G_{CO2}$) (**a**) and two phases ($L_{CO2}$ + $G_{CO2}$) (**b**); (**c**) B-type fluid inclusions with three phases (B1-type) (c) and two phases (B2-type) (**d**); (**e**) C1- (the upper) and C2-type (the lower) fluid inclusions; (**f**) D-type fluid inclusions.

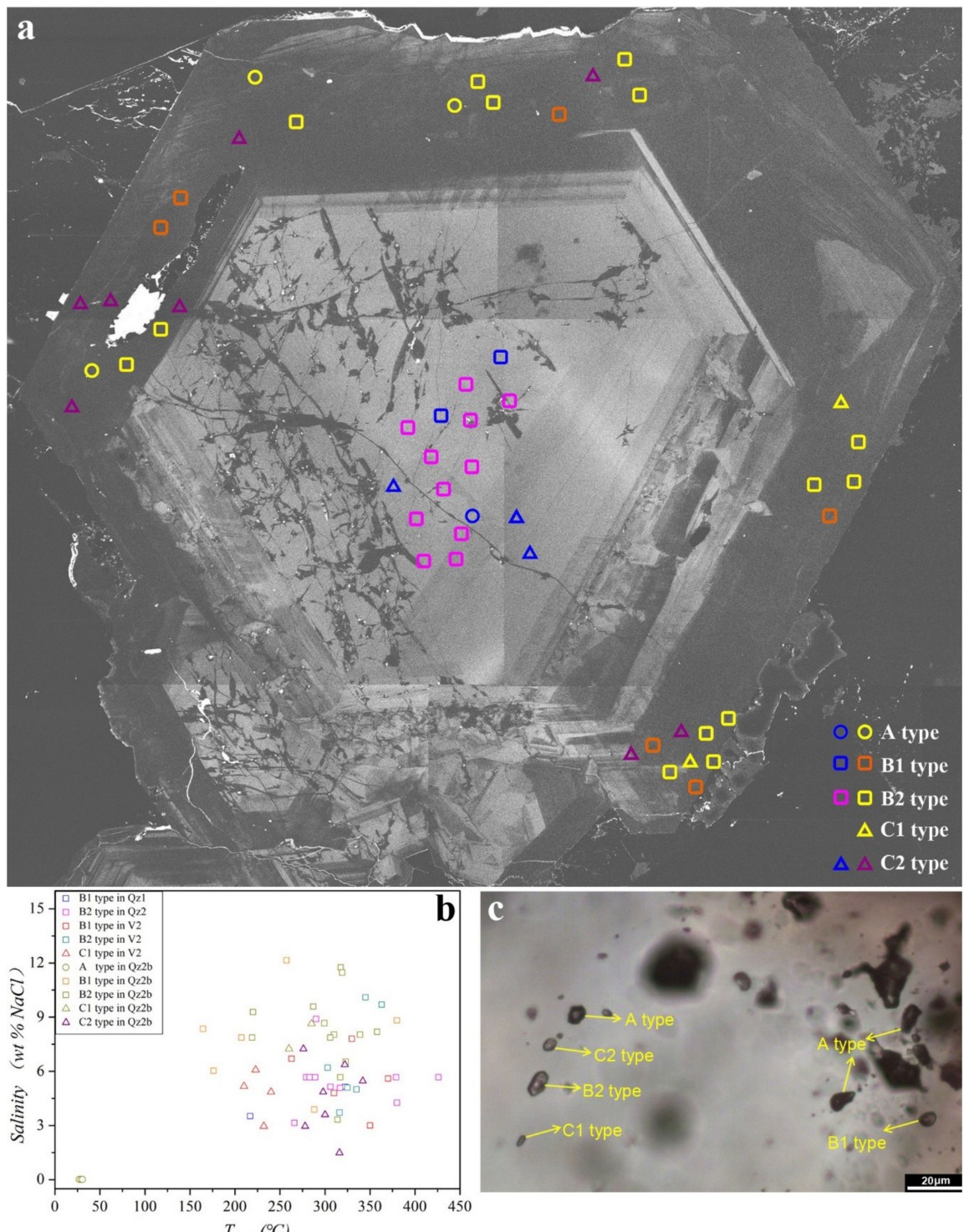

**Figure 7.** SEM-CL imaging of quartz with distributions of fluid inclusions (**a**) and a plot of homogenization temperatures and the salinity of different types of inclusions (**b**) revealed by CL image in (**a**). Due to a lack of Qz2a in (**a**), fluid inclusion data are from Qz2a bearing V2 veins. (**c**) Boiling fluid inclusions assemblage in Qz2b.

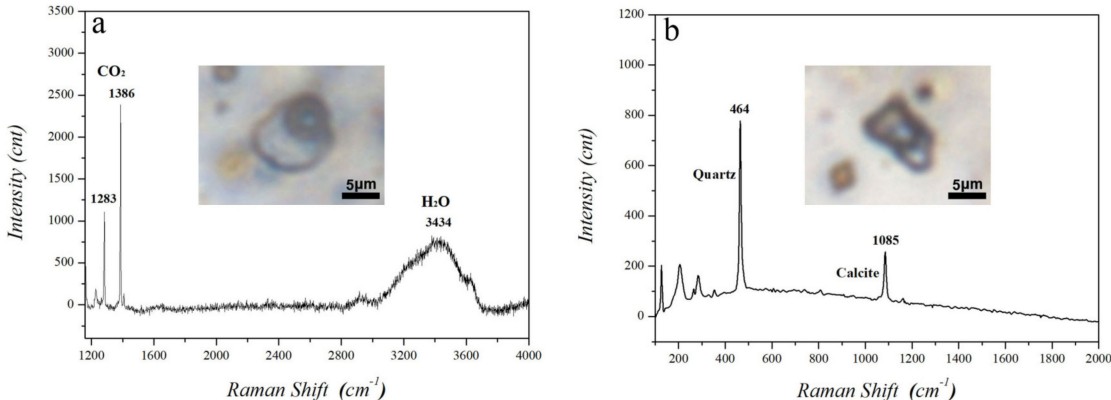

**Figure 8.** Raman spectra of B1-type fluid inclusion (**a**), and D-type fluid inclusion fluid inclusions at the Linglong gold deposit (**b**).

During cooling and heating runs, A-type inclusions showed carbonic phase melting at around −56.6 °C, indicating almost pure $CO_2$; clathrates were seldomly observed, indicating low concentrations of water; and the inclusion homogenized between 27–30.9 °C to gas phase; B1-type inclusions showed carbonic phase melting at −56.6–−58.3 °C; clathrate melting occurred at 2.7–8.9 °C, corresponding to 2.8–12.3 wt% NaCl and final homogenization temperatures occurred at 260–330 °C to liquid or gas phase; B2-type inclusions showed carbonic phase melting at −56.6–−58.7 °C; clathrate melting at 2.8–10.0 °C, corresponding to 0 wt%–12.2 wt% NaCl and final homogenization occurred at 202–340 °C to liquid or gas phase; C1-type inclusions showed ice melting at −5.6–−4.6 °C, corresponding to 7.3 wt%–8.7 wt% NaCl, and final homogenization occurred at around 270 °C to liquid phase; C2-type inclusions showed ice melting at −8–0 °C, corresponding to 0.2 wt%–11.7 wt% NaCl, and final homogenization occurred at 120–280 °C to liquid phase; D-type inclusions showed clathrate melting occurred at 2.7–7.0 °C, corresponding to 2.8 wt%–12.28 wt% NaCl, and homogenized at 210–315 °C to liquid phase. The microthermometric results of all fluid inclusions are listed the Table 2, but only those with CL images are plotted in the Figure 7b.

**Table 2.** Microthermometric results of fluid inclusions at Linglong gold deposit.

| Sample No. | Vein | FI Type | Inclusion Number | $T_{m,CO2}$ (°C) | $T_{m,ice}$ (°C) | $T_{m,clath}$ (°C) | $T_{h,CO2}$ (°C) | $T_{h,tot}$ (°C) | Salinity (wt%) | $CO_2$ Density (g/cm$^3$) | Bulk Density (g/cm$^3$) |
|---|---|---|---|---|---|---|---|---|---|---|---|
| 17LL21 | V1 | A, | 1 | −57.0 | / | / | / | 27.5 (G) | / | 0.28 | / |
| | | B1, | 4 | −57.2~−57.0 | / | 2.7~8.6 | 29.7~30.9 (L) | 299.6 (av.) (L or G) | 2.76~12.28 | 0.53~0.60 | 0.72~1.00 |
| | | B2, | 10 | −58.7~−57.4 | / | 4.5~9.0 | 28.2~30.9 (L) | 317 (av.) (L) | 2.0~9.74 | 0.53~0.65 | 0.81~0.98 |
| | | D | 1 | / | / | 2.7 | / | 208.2 (L) | 12.28 | 0.53 | 0.92 |
| 10LL04 | V2 | B2, | 1 | −57.0 | / | 5.6 | / | 269.5 (L) | 9.14 | / | / |
| | | C2 | 2 | / | −1.5, −1.0 | / | / | 282.9 (av.) (L) | 1.65, 2.47 | / | 0.75, 0.76 |
| 17LL01 | V2 | B1, | 6 | −57.4~−57.1 | / | 4.2~8.5 | 26.95~30. 9 (G) | 329.5 (av.) (L or G) | 2.92~5.60 | 0.35~0.71 | 0.79~0.90 |
| | | B2, | 12 | −57.5~−57.0 | / | 3.6~8.1 | 27 (L) | 331.5 (av.) (L or G) | 2.96~10.19 | 0.25~0.68 | 0.79~0.86 |
| | | C2 | 6 | / | −2.2~0.1 | 4.2~8.5 | / | 263.4 (av.) (L) | 0.17~3.60 | / | 0.69~0.95 |
| 17LL05 | V2 | B1, | 3 | −56.5~−56.4 | / | 6.0~6.9 | 26.3~30.8 (G) | 320.6 (av.) (L) | 6.02~7.54 | / | 0.73~0.94 |
| | | B2, | 4 | −57.1~−56.3 | / | 4.5~9.5 | 30.8 (G) | 341.5 (av.) (L) | 5.50~7.91 | 0.27~0.39 | 0.74~0.82 |
| | | C2, | 2 | / | −4.0, −2.2 | / | / | 222.6, 235.9 (L) | 3.60~6.37 | / | 0.85~0.96 |
| | | D | 1 | / | / | 7.0 | / | 315.0 (L) | 5.68 | / | / |
| 17LL09 | V2 | B2, | 4 | −57.8~−57.6 | / | 4.0~8.3 | 25.3~29.0 (L) | 321.7 (av.) (L) | 3.33~10.48 | 0.63~0.71 | 0.92~0.96 |
| 17LL13 | V3 | A, | 3 | −56.8~−56.7 | / | 4.7~7.0 | / | 27.0~30.5 (G) | 3.89~8.35 | 0.27~0.37 | / |
| | | B1, | 8 | −58.3~−56.7 | / | 2.8~8.9 | 24.9~30.9 (G) | 263.1(av.) (L or G) | 0.02~12.15 | 0.53~0.76 | 0.74~0.94 |
| | | B2, | 81 | −57.7~−56.7 | / | 2.8~10.0 | / | 287.1 (av.) (L or G) | | 0.53~0.64 | 0.87~0.95 |
| | | C1 | 2 | / | −5.6~−4.6 | / | / | 270 (av.) (L) | 7.25~8.65 | / | 0.97~0.98 |
| | | C2, | 18 | / | −8.0~−0.5 | / | / | 255.3 (av.) (L) | 0.83~11.70 | / | 0.60~1.0 |
| | | D | 1 | / | / | / | / | 220.0 (L) | / | / | / |
| 17LL15 | V4 | A, | 2 | −56.7~−56.6 | / | / | / | 29.9, 30.9 (L) | / | 0.60, 0.59 | / |
| | | B2, | 14 | −56.7~−56.6 | / | 6.7~8.1 | 29.7~30.8 (G) | 201.7(av.) (L) | 3.7~6.2 | 0.33~0.40 | 0.65~0.81 |
| | | C2 | 2 | / | −8.0~−7.1 | / | / | 115.6, 143.1 (L) | 10.6~11.7 | / | 1.0~1.03 |
| 12LL11 | V4 | C2 | 7 | / | −7.8~−2.3 | / | / | 144.0 (av.) | 3.76~11.46 | / | 0.89~1.04 |

Note "/" denotes no data available.

### 4.3. Quartz Solubility Diagrams

Two types of solubility diagrams are constructed to study the effect of different parameters. P-T diagrams are constructed at constant fluid compositions (salinity: 5 wt% NaCl; $X_{CO2}$: 0.1 and 0.15) to investigate the effects of pressure and temperature (Figure 9). Figure 10 shows solubility versus temperature diagrams (Figure 10) at constant pressures, to visualize the effects of salinity and carbon dioxide contents.

In the P-T diagram (Figure 9), it is evident that solubility isopleths can be divided into three fields, i.e., T-sensitive yet P-inert field where isopleths are nearly vertical (TsPi); P-sensitive yet T-inert field where isopleths are nearly horizontal (PsTi); and retrograde solubility field where isopleths concave down towards (ReSo). The TsPi field occupies the high-pressure region, and its boundary with the other two field show a linear increase between P and T. The PsTi field occupies intermediate pressure and temperature region, while the ReSo field occupies low pressure region. It is noteworthy that the solubility isopleths show no discontinuity across the phase boundaries. In the TsPi field, the spacing between isopleths gradually increase with decreasing temperature. By contrast, spacing between the isopleths in PsTi field decreases with decreasing pressure, but at lower magnitudes than those for TsPi. It is noted that, in the studied temperature range, quartz solubilities are much lower at low pressures than those at high pressures. An increase of $CO_2$ molar content from 10% to 15% does not change the general topology, but shifts the temperature and pressure covers of the regions of TsPi, PsTi, and ReSo.

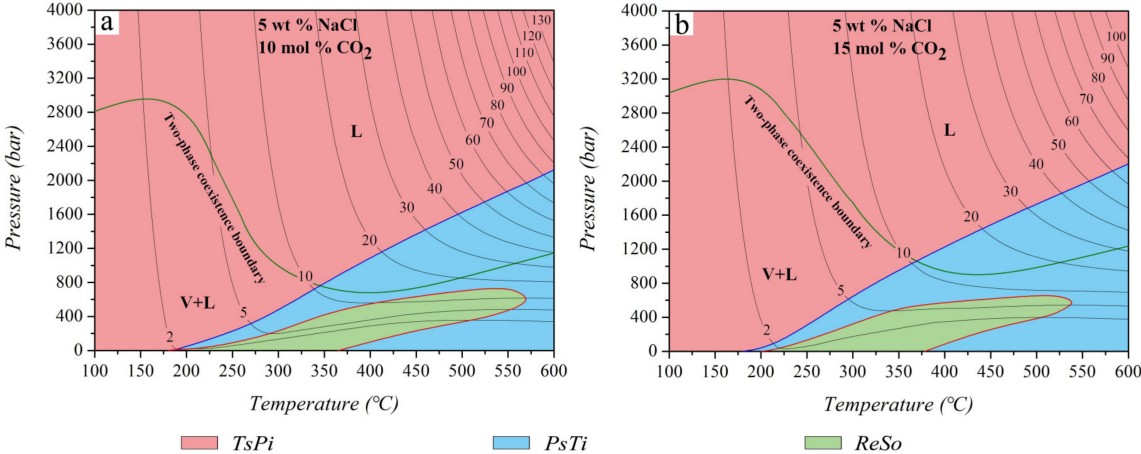

**Figure 9.** P-T diagram showing the single phase (liquid) and two-phase (vapor + liquid) field of $H_2O$-NaCl-$CO_2$ ternary system, containing 10 mol% $CO_2$ (**a**) and 15 mol% $CO_2$ (**b**), contoured with quartz solubility isopleths. Three regions exhibiting different dependence on P and T are estimated (TsPi: temperature sensitive pressure inert; PsTi, and ReSo). For a fixed composition, if $(T_0, P_0)$ is any point on the curve of Equation (1), the boundary between TsPi and PsTi (the blue curve) is the locus of points of isopleths that have $(\partial m/\partial T)_{P=P_0} = (\partial m/\partial P)_{T=T_0}$; the boundary between PsTi and ReSo (the red curve) is the locus of points of isopleths that have a slope of zero.

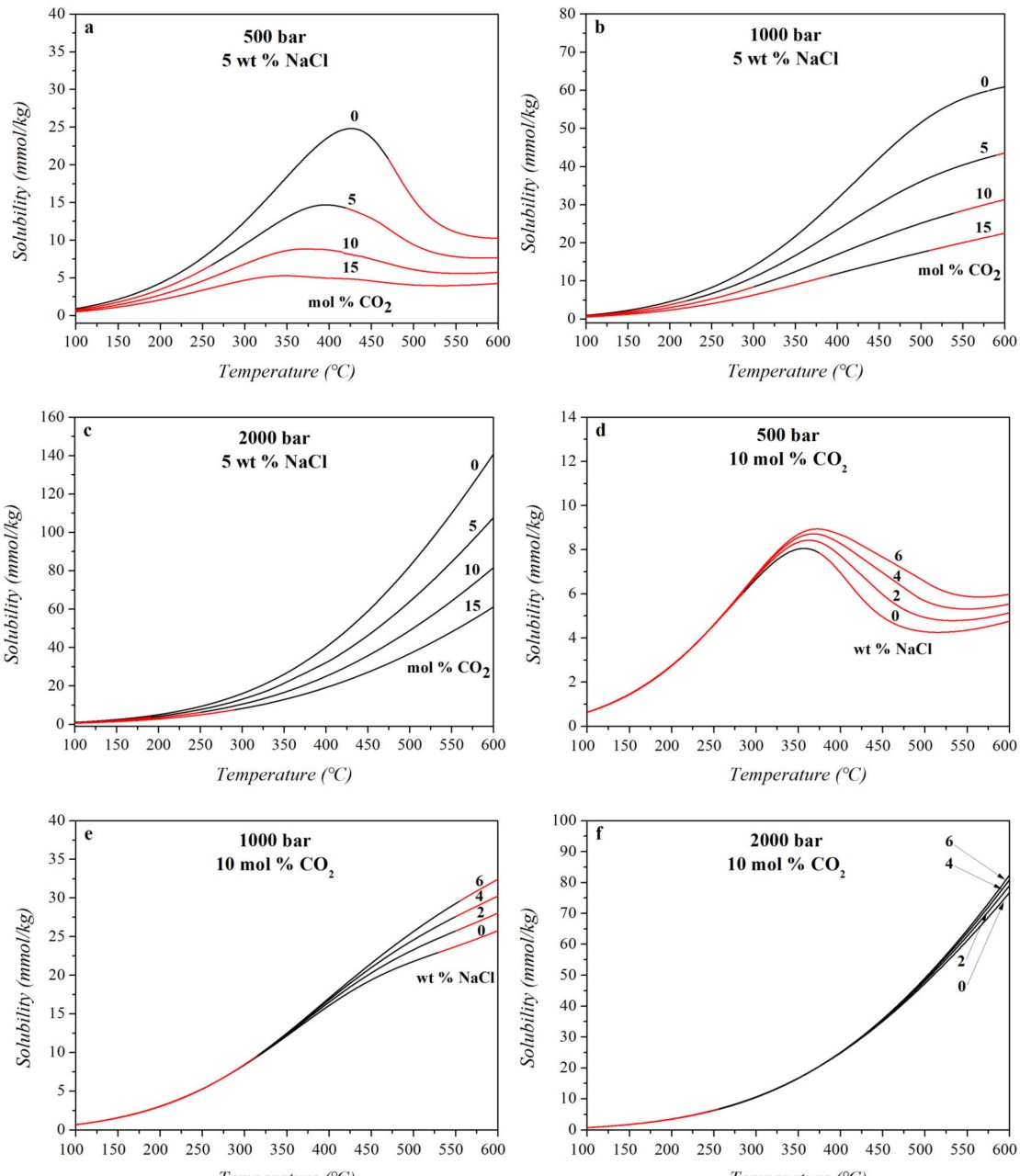

**Figure 10.** Predictions of quartz solubility in $H_2O$-$CO_2$-NaCl hydrothermal solutions. (**a–c**) are the fluids containing of 5 wt% NaCl and 0 mol%–15 mol% $CO_2$ compositions at 500, 1000, and 2000 bar conditions respectively; (**d–f**) are the fluids containing of 10 mol% $CO_2$ and 0 wt%–6 wt% NaCl compositions at 500, 1000 and 2000 bar conditions respectively; Notes: The computing results were obtained from AD model (2009). Black lines represent quartz solubility in liquid phase of hydrothermal fluids, and the red lines are quartz solubility in vapor + liquid phase of hydrothermal fluids. Quartz solubility in millimole $SiO_2$ per kilogram water.

To investigate the influence of $X_{CO2}$ and salinity, solubility versus temperature diagrams with three constant pressures (500 bar, 1000 bar, and 2000 bar), representing interception with different fields defined in Figure 10, were constructed. $CO_2$ has a negative effect on quartz solubility regardless of pressure and temperature (Figure 10a–c), due to its non-electrolyte nature that reduces silica hydration degree [63]. As Figure 10a–c depict, for a fluid with fixed salinity, the addition of $CO_2$ always reduces quartz solubility at 500, 1000 and 2000 bar conditions. In addition, the magnitude

of solubility decrease among different $X_{CO2}$ dramatically reduces with decreasing temperatures, but increases with decreasing pressures. For instance, an increase of $X_{CO2}$% from 5% to 10% at 500 °C and 1000 bar can reduce quartz solubility by 10 mmol/kg (from 35 mmol/kg to 25 mmol/kg); whereas the magnitude in quartz solubility change drops to around 2.5 mmol/kg at 300 °C and 11 mmol/kg at 2000 bar. At 500 bar (Figure 10a), solutions with different $X_{CO2}$% (between 0% mol and 15% mol) exhibit retrograde solubility between 600 °C and 350 °C. The magnitude of retrogression decreases with increasing $X_{CO2}$. The peak point of quartz solubility shifts consistently to lower temperature with increasing $X_{CO2}$. At 1000 bar and 2000 bar, no retrograde solubility is present, and instead, solubility decreases monotonically (Figure 10b,c). It should be noted that the observed retrograde solubility patterns are a common feature for low pressures, not limited to 500 bar.

As to the effect of the electrolyte of NaCl, it demonstrates a complex effect on quartz solubility by affecting the solution volume and changes hydration. At low pressure (500 bar) and the temperature approaching the critical point of water, the quartz solubility demonstrates a "salting-in" (quartz solubility enhanced when $X_{NaCl}$ is increasing) effect and some extent of retrograde phenomenon (Figure 10d). At higher pressures, the "salting-in" effect of quartz solubility diminished progressively and the retrograde phenomenon is invisible (Figure 10e,f). As density models depict [63–65], at higher pressure conditions (>2000 bar), the "salting-in" effect of quartz solubility will all changed into the "salting-out" (quartz solubility reduced when $X_{NaCl}$ is increasing) effect, because of the expanding of fluid molar volumes, thereby increasing the average distance between water volumes and decreasing quartz solubility. Similar to the $CO_2$ effect, magnitude of solubility decrease among different $X_{NaCl}$ dramatically reduces with decreasing temperatures, but increases with decreasing pressures. However, the magnitude of solubility change is significantly lower than that of the $CO_2$ effect (Figure 10d–f). Similarly, at low pressures (e.g., 500 bar), solution with 10% mol $CO_2$ and variable salinity displays retrograde solubility in the temperature range of 550–350 °C, and the peak point of solubility shifts to lower temperatures with decreasing salinity (Figure 10d). At higher pressures (e.g., 1000 bar and 2000 bar, Figure 10e,f), quartz solubility decreases monotonically.

## 5. Discussion

### 5.1. Fluid Temperature and Pressure Estimated from Fluid Inclusions

Fluid inclusions have been used as a common means of estimating fluid pressure and temperatures. Normally, with phase-transition temperatures (e.g., $T_h$, $T_{m,ice}$) and appropriate PVTX diagrams or equations of state, minimum trapping temperature and pressure can be obtained. To know the exact trapping conditions, either independent P or T estimate such as those from mineral thermobarometers, or the estimates of rock load should be applied [68]. In the case of immiscible fluid inclusion assemblage (the coexistence of two fluids with contrasting density and composition), actual trapping conditions can be determined by intercept of isochores from the two types of inclusions.

In Qz1 of the Linglong goldfield, primary fluid inclusions are characterized by B-type liquid-vapor $H_2O$-$NaCl$-$CO_2$ inclusions (Figure 7a) that were trapped in the single-phase field of a P-T diagram. Trapping conditions of this type of inclusion is estimated with independent pressure constraints for the Jiaodong region. Liu et al. [45] studied the exhumation history of the Sanshandao gold deposit in the western part of the Jiaodong gold province, by an integrated method of structural analysis and (U-Th)/He thermochronology, and they concluded that the deposit was eroded at least 5 km. A similar result has been obtained by Li et al. [69], which suggested that the Daliuhang gold deposit in the eastern part of the Jiaodong gold province may have exhumed 5 km since its formation, by employing Al-in-hornblende barometer and geochronology. These results may point to comparable exhumation level in the entire province. Assuming the Linglong goldfield, located between the above mentioned deposits, has experienced a similar exhumation magnitude, the intercepts between lithostatic pressure at mineralization timing (2.0–2.6 kbar) of Li et al. [69] and inclusion isochores of B1-type inclusions in Qz1 yield the trapping temperatures (293–536 °C, the grey field in Figure 11a). This large temperature

range might be attributed to uncertainty concerning if B1- or B2-type represents primary inclusions in Qz1. However, it is more likely that B1-type inclusions represent primary inclusions, since B2-type inclusions are commonly presented in Qz2b. Taking isochores of B1-type inclusions; the temperatures are narrowed to 388–502 °C (the dark field in Figure 11a).

In Qz2b, the immiscible assemblage of fluid inclusions are present (coexistence of A- and C1-type inclusions), providing an opportunity to constrain the trapping conditions. The isochores of A and C1 inclusions are constructed, and their intercepts are taken as the estimates of trapping temperatures (292–331 °C) and pressures (273–431 bar) (Figure 11b). For Qz2a, due to a lack of CL-aided inclusion data, we tentatively infer that they were formed at similar temperature but higher pressures (300–350 °C and 0.5–1.0 kbar).

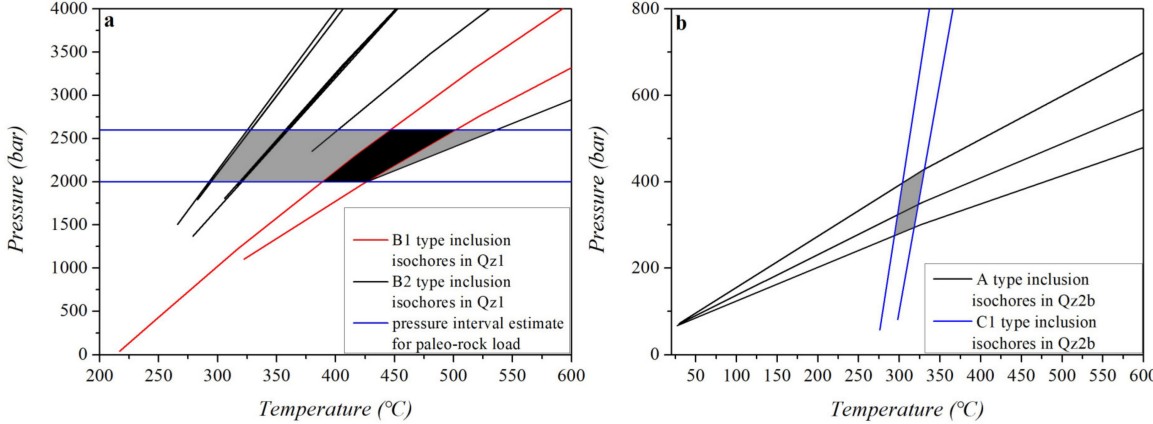

**Figure 11.** Estimated trapping pressures and temperatures by fluid inclusion isochores. (**a**) Interception between B-type inclusion isochores and pressure estimated from paleo-rock load for Qz1; (**b**) Interception between A- and C1-type inclusion isochores for Qz2b.

## 5.2. Quartz Deposition and Dissolution Events at Linglong

Quartz deposition events at Linglong have been intensively studied [26,70]. Three major quartz deposition events have been proposed based on macroscopic and microscopic observations of three generations of quartz-bearing veins, i.e., early barren quartz vein (V1 defined in this study), followed by pyrite quartz (V2) and polymetallic sulfide quartz (V3) veins. The same classification scheme has also been put forward for most gold deposits in the Jiaodong gold province [71]. A potential pitfall of using vein sequences to represent quartz sequence is that quartz of different generations might not be discernible by conventional observations [8]. With the aid of CL imaging, three generations of quartz were recognized for the Linglong deposit, which is consistent with the vein chronology. Detailed microtextural features are uncovered for each quartz deposition. Early quartz (Qz1) deposited during V1 quartz vein formation is characterized by relatively euhedral morphology (Figure 3a), bright CL and weak concentric zonings (Figure 12), likely suggesting deposition at relatively stable physico-chemical conditions (such as high temperature and slightly fluctuated pressure [72–74]). CL-aided fluid inclusion studies suggest that these quartz were deposited from a single-phase carbonic aqueous solution at temperatures of 388–502 °C and pressures of 2.0–2.6 kbar. These quartz grains are commonly highly fractured and even brecciated, likely indicating that the early-formed V1 veins might have been reopened and served as conduits and depositional sites for later fluid events. During the course of Qz1 deposition, little metal sulfide and gold was precipitated.

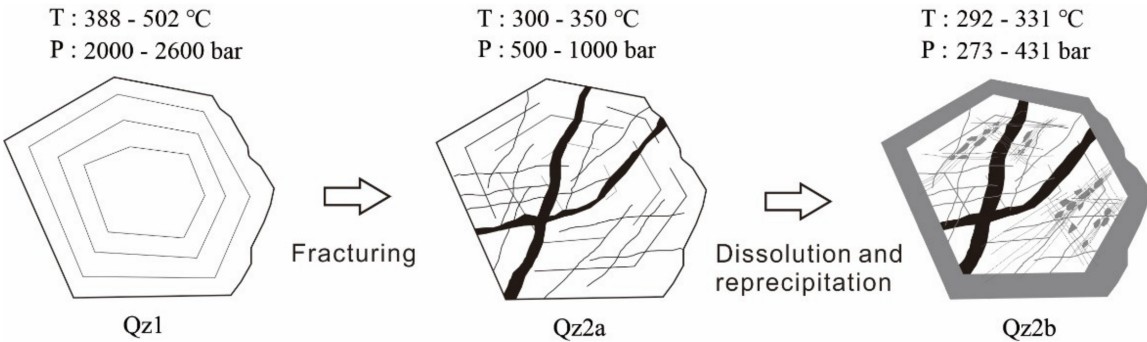

**Figure 12.** A sketch map of three different types of quartz in vein generations of Linglong goldfield.

The following quartz deposition events are represented by the formation of Qz2a and Qz2b. A significant difference revealed by the CL study is that the amount of quartz is much more limited than that observed with conventional methods. More importantly, intimate and reliable spatial relations between these two quartz generations with metal sulfides and gold are identified. The dark quartz CL signature might indicate formation under conditions which are completely different from those of Qz1 (Figure 12), probably lower temperatures, although evidence of trace elements are needed to make a quantitative conclusion [72–74]. Qz2a and Qz2b have no distinctive CL features, except for the style of depositional site (Qz2a fills in brittle fractures, whereas Qz2b pseudomorphically replaces Qz1) and metal sulfide associations (Qz2a mainly intergrow with pyrite and possibly invisible gold, whereas Qz2b mainly coexist with polymetallic sulfides and visible gold). This might be an indication of slight yet significant changes in fluid properties and physico-chemical conditions. CL-aided fluid inclusion studies revealed that Qz2b was precipitated from immiscible fluids characterized by $CO_2$-rich vapor, $H_2O$-rich liquid and mixtures of the two at temperatures of 292–331 °C and pressures of 0.27–0.43 kbar.

Besides quartz deposition events, a period of quartz dissolution is recognized through CL studies, which is the secondary CL texture of "splatters" and "cobweb" and pseudomorphic replacement in Qz1 (Figure 5a,c,e,g). The former texture in early quartz of porphyry Cu systems have been interpreted as a result of dissolution [8], while the latter might a result of combination of dissolution and reprecipitation. The dissolution event was immediately followed by the deposition of Qz2b, which probably means that the dissolution process had created spaces for the precipitation of quartz, sulfide, gold and other minerals.

*5.3. Mechanism of Early Quartz Deposition*

CL-aided fluid inclusion petrography and microthermometry suggests that the early quartz (Qz1) was deposited at temperatures of 388–502 °C and pressures of 2.0–2.6 kbar, with minor amounts of pyrite, but little gold. The pressure temperature range indicates precipitation in the TsPi field of the Figure 9. In this field, pressure is a quite inefficient factor for changing quartz solubility. Salinity is also not efficient for controlling quartz solubility, especially at high pressures, as demonstrated in Section 4.3 (Figure 10f). Therefore, among the parameters that may affect quartz solubility, temperature and $CO_2$ content may be decisive. Assuming an original transporting fluid (10 mol% $CO_2$, 5 wt% NaCl) that carries $SiO_2$ at the solubility of 90–130 mmol/kg, at 550–600 °C and 3–3.5 kbar, decompressional cooling to the depositional conditions of Qz1 can precipitate 70% to 80% silica, and this massive quartz precipitation can be further enhanced to over 90 wt% if $CO_2$ content is increased by 5 mol% (Figures 9 and 13).

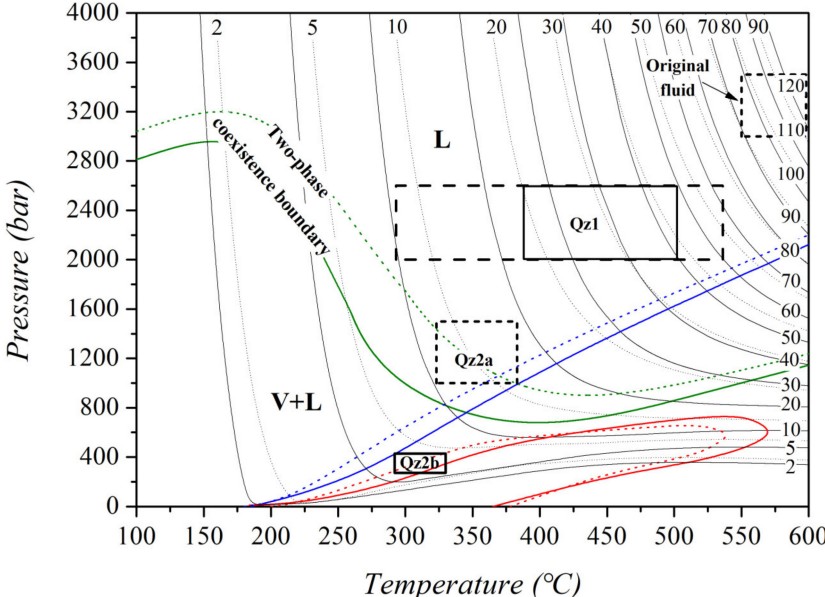

**Figure 13.** Formation of different veins in Linglong gold deposit. Notes: The black solid curves and dash curves represent quartz solubility isopleths in liquid and vapor + liquid hydrothermal fluids, containing 10 mol% $CO_2$ and 5 wt% NaCl and 15 mol% $CO_2$ and 5 wt% NaCl solute, respectively. The green solid curves and dash curves, the blue solid curves and dash curves, and the red solid curves and dash curves are the boundaries of liquid phase and the vapor + liquid phase, of the TsPi and the PsTi, and of the PsTi and ReSo in different hydrothermal fluids, containing 10 mol% $CO_2$ and 5 wt% NaCl and 15 mol% $CO_2$ and 5 wt% NaCl, respectively. Solid squares show the calculated conditions of Qz1 and Qz2b, while dashed squares show the assumed conditions of original fluids, Qz1 and Qz2a.

However, before attributing Qz1 quartz deposition to temperature, it is imperative to justify how cooling occurred, and to what extent. Numerical considerations for ore formation have been undertaken by Toulmin and Mark [75] and Rose [76], whose conclusions may be used to interpret the case of Linglong. Rose [76] developed a model to investigate the effects of cooling $2.5 \times 10^9$ metric tons of solutions in a 0.5 km× 1 km × 1 km rock volume at different fluid flow velocities from 700 to 400 °C for the formation of porphyry—style ore body. The conclusion reached was that simple cooling by conduction to wallrocks seemed to not be tenable. Instead, adiabatic cooling (the so-called throttling) and/or mixing with cool groundwater should be necessitated. This model works perfectly for the case of Linglong, where the mass of solution involved ($2.25 \times 10^{10}$ metric tons, unpublished data), dimension of rock volume (0.01 km × 1 km × 5 km) and temperature profile for an averaged quartz vein are comparable to those modeled parameters. Considering this, it is reckoned that cooling by conduction was not effective for Linglong, either. Moreover, adiabatic cooling is also considered impossible for Linglong, since no evidence of dramatic pressure drop such as breccia has been observed. Similarly, mixing with cool groundwater is also considered unsuitable, because existing H-O isotopic studies consistently revealed the dominance of magmatic components without meteoric involvement [26,27]. In addition, solubility models for pyrite and gold in $H_2O$—10 wt% NaCl solution suggests that temperature has a strong negative effect on metal sulfides and gold (Figure 14a of Kouzmanov and Pokrovski [77]), and thus, cooling will inevitably lead to precipitation of sulfides and gold, which is contrary to the observations at Linglong. Therefore, the above discussions on the thermal aspect and solubility behaviors of sulfides and gold may serve as arguments against the major role of cooling for the early veining at Linglong.

Similarly, the possibility of the $CO_2$ effect should also be evaluated. At Linglong, formation of Qz1 occurred along with alkaline metasomatism that significantly increased the amounts of secondary biotite. This formation of biotite may have preferentially consumed $H_2O$ but exclude $CO_2$ and have led

to increased $CO_2/H_2O$ ratios, which is essentially $CO_2$ content in the fluid. As is shown in Figure 13, an increase in $CO_2$ content will reduce quartz solubility, and therefore, it is likely that the strong biotitization of wall rocks may have contributed to the decrease of quartz solubility and thus quartz deposition. However, there are a lack of data on the scale of biotitization and thus, it is impossible to quantitatively estimate $CO_2$ content change. If the acidity of the fluid system is internally buffered by $CO_2$ through the chemical reaction:

$$CO_2 + H_2O = HCO_3^- + H^+ \tag{1}$$

as suggested by Phillips and Powell [78] and Phillips and Evans [79,80], biotitization will consume $OH^-$ and thus reduce fluid pH value. The change in fluid acidity may explain the reversal behavior between quartz and pyrite/gold. As is shown by the solubility models of Kouzmanov and Pokrovski ([77] and Figure 14c therein), a reduction in pH will increase the solubility of pyrite and, to some extent, gold. Therefore, the biotitization (pH increase) may have been responsible for the deposition of Qz1 without pyrite and gold in the early veining process.

*5.4. Mechanism of Late Quartz Deposition*

CL petrography has revealed two prominent features for Qz2a, i.e., occurrence in fractures that, in places, brecciated Qz1a; and significantly reduced amount in comparison to Qz1. The first feature indicates that the fluids may have experienced overpressure before the deposition of Qz2a. Fluid inclusion data of this study, although not as reliable as those for Qz1 and Qz2b obtained with the aid of CL, indicate that Qz2a was likely deposited at lower temperatures in the two-phase field and thus lowered quartz solubility (Figure 7). If the source remains the one as to that of Qz1, decompression alone would have caused quartz deposition in a scale similar to the early veining. It is plausible that the second feature argues for an either significantly cooled source or mixing with a silica—undersaturated fluid. The latter hypothesis is supported by H–O isotopic data, which revealed the presence of meteoric water during the formation of pyrite quartz veins (V2 of this study) [27]. Ingression of the silica—undersaturated meteoric water may have diluted the ore fluids and limited the amounts of precipitated quartz. Meanwhile, enhanced cooling induced by the meteoric ingression may have been responsible for the deposition of large quantities of sulfides and gold (Figure 14a of Kouzmanov and Pokrovski [77]).

Because no distinctive difference in CL and chronological evidence are present for Qz2a and Qz2b, it is highly likely they represent a continuous fluid evolution. A marked feature of the Qz2b formation process is that it may have undergone quartz dissolution prior to reprecipitation, as evidenced by the "splatter" and pseudomorphic replacement textures. According to the solubility behavior of quartz, dissolution would require: (1) heating in the TsPi region; (2) compression in the PsTi field; or (3) cooling through ReSo field. Fluid inclusion microthermometric data revealed that Qz2b might have been deposited at temperatures of 292–331 °C and pressures of 273–431 bar, corresponding to an overlapping region between PsTi and ReSo fields (Figure 13), ruling out the first hypothesis. However, no pressurization events have been been suggested for Linglong, and thus the second hypothesis is considered impossible. Therefore, it is reasonable to infer that the quartz dissolution prior to Qz2b is considered as a result of cooling in the ReSo field of quartz solubility. In addition, $CO_2$ decrease may also be significant in increased quartz solubility. This process may have been due to vapor loss in the two-phase field (such as the model proposed by Hu et al. [71]) in the presence of fluid boiling in an open system, although the degree of this effect cannot be evaluated, due to a lack of relevant data. More importantly, the decrease in $CO_2$ may have been caused by phyllic alterations with consume $H^+$ and thus move the Equation (1) to the right side, as opposed to biotitization. Qz2b, along with polymetallic sulfides and gold, deposition may have been induced by further cooling. To this end, a conclusion can be drawn that fluid cooling in ReSo field of quartz solubility combined with $CO_2$ loss might have been responsible for quartz dissolution, and continued cooling may have resulted in the late veining at Linglong.

## 6. Conclusions and Implications

Key findings of this study bear important implications for understanding genetic processes of gold veins in the Jiaodong gold province and other orogenic terrains at large. In Jiaodong, gold mineralization in most deposits, regardless of ore styles, occurred after a preceding major episode of Au-barren quartz deposition, as thick quartz veins or as thin quartz veinlets [81]. The mechanism of this massive quartz depositional event and its role in the following deposition of metal sulfides and gold remains largely unattended. Quartz solubility interpretations, coupled with CL-aided fluid inclusion analyses, revealed an insignificant role of decompression due to "near vertical" isopleths and a limited role of fluid cooling, regarding the efficiency of heat loss in the formation of this early quartz veining event. The exclusion of pressure, especially huge pressure dropping induced by earthquakes [17], as a major factor for quartz deposition, is based on quartz solubility analysis in carbonic brine solutions, which seems to be more accurate than those based on pure water. In the meantime, an increase in the content of $CO_2$ in fluid is highlighted, due to its capacity of decreasing quartz solubility in the relevant physico-chemical conditions. This interpretation helps to establish a consequential link between quartz veining and the coeval biotitization that may have changed fluid pH, and provides a possible explanation for the Au-barren nature of this early veining event.

Another discovery made by this study is that the quantities of quartz that are genetically related to sulfides and gold is much lower than expected by most stable isotopic and fluid inclusion studies that have been based on conventional petrographic methods other than CL. This discovery should bring attention to future studies of gold deposits in orogenic intrusion-related settings, to employ in situ analytical techniques for obtaining stable isotopic and fluid inclusion data and to interpret the data on a firm basis of CL petrography. Such practice will significantly improve the quality of data and interpretations on fluid source and evolution, which are currently problematic due to fluid overprints [39]. In addition, CL microtextures suggest that the early quartz veins are very important site for later deposition of sulfides and gold, as evidenced by the fracturing, brecciation and replacement textures. In this sense, the early veining processes are also an important part of the entire fluid process and have a role to play for the gold genesis.

The roles of temperatures, pressures, phase separation, fluid-rock interaction and fluid mixing have been intensively discussed by previous studies of gold complexing, transport and deposition [19,79,80]. This study offers a new perspective for addressing this issue with quartz solubility analysis. Detailed discussion on quartz dissolution and precipitation process for the main stages of gold formation indicates the important roles of fluid mixing and fluid-rock interaction in the quartz veining process. These interpretations are aligned with the previous aforementioned studies.

**Author Contributions:** Conceptualization, Q.W. and X.L.; methodology, Q.W. and X.L.; validation, H.F. and X.L; analytical analysis, Q.W.; resources, Q.W., H.F., J.P. and X.L.; writing—original draft preparation, Q.W. and X.L.; writing—review and editing, H.F. and J.P.; funding acquisition, Q.W., H.F., L.L., J.P. and X.L. All authors have read and agreed to the published version of the manuscript.

**Funding:** This research was funded by the National Natural Science Foundation of China [Nos. 41602073, 41802084], the State Key Laboratory of Ore Deposit Geochemistry Foundation of Institute of Geochemistry, Chinese Academy of Sciences [No. 201707] and Extra & Co project (Valorisation-Instituts Carnot n°ANR-15-CNRT-003).

**Acknowledgments:** Alexandre Tarantola and Weiping Deng are acknowledged for providing suggestion and optimizing code on phase equilibrium and quartz solubility calculations. Tingguang Lan is appreciated for help with the field work. We also thank Shaohua Dong and Liangliang Huang for assistance with SEM-CL and Raman sessions. Assistant Editor Evelyn Wang is acknowledged for inviting X.L. to write this paper. Two anonymous reviewers are thanked for their critical and constructive remarks.

**Conflicts of Interest:** The authors declare no conflict of interest.

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
