# Peer review of "Auriferous Quartz Veining Due to CO2 Content Variations and Decompressional Cooling, Revealed by Quartz Solubility, SEM-CL and Fluid Inclusion Analyses (The Linglong Goldfield, Jiaodong)"

_minerals, doi:10.3390/min10050417_

Round 1

Reviewer 1 Report

The manuscript 'Auriferous quartz veining due to decompressional cooling and CO2 content variations revealed by quartz solubility, SEM-CL and fluid inclusion analyses (the Linglong goldfield)' is a very interesting contribution. 

I am attaching the edited manuscript, which reflects my concerns. The text needs to be revised and proof-read by a native English speaker and the thermodynamic modeling methods section needs to be expanded significantly. How were the phase diagrams calculated? What were the input species and paramters etc.? This needs significant improvement. 

The title and abstract seem to be misleading. The discussion and conclusion state that the first quartz vein generation was formed by a reaction that formed biotite and lead to quartz precipitation (does the mass balance for that work? Can we see pictures?), but the title and abstract say quartz vein formation through decompressional cooling - which was ruled out in the discussion. 

I did some English annotations, but not to the extend necessary. 

Reviewer 2 Report

General Comments:

A) Very interesting work. Thanks for the hard work.

B) It would have been more beneficial if the pictures were larger in size.

C) Some documentation is required for demonstrating mineral assemblages (Maybe a few optical mineralogy pictures). In particular, it is not clear if the gold found in V2 and V3 veins are in chemical equilibrium with quartz. Pictures needed to demonstrate these associations.

D) Pictures required to demonstrate the primary/secondary origin of fluid inclusions. Specially for the boiling assemblage in Qz2b.

E) The manuscript needs to be checked for grammar. I have mentioned some errors, but English is not my native language so there might be more.

LINE 35: 

Because of its abundance and properties, quartz and silica minerals:

 Delete: and silica minerals. Your sentence is singular.

LINE 36: 

Silicate Minerals? 

  LINE 41-44: You'd have to be more clear of what you mean by "veining processes is well understood". And how it is well understood by for example oxygen isotopic composition, CL, and so forth.

LINE 47-48:

Reference?

LINE 54:

Mention where your deposit is located.

LINE 55:

Vein-type gold deposit (remove “s”)

LINE 63-68:

This is not geology. Should have it’s own section.

Fig 1: Where is North?

Fig 5i: image quality?

LINE 266:

Grammar: Change "C type inclusions are normally consisted of” to "C type inclusions normally consist of”

LINE 277-279:

Picture required to demonstrate boiling assemblage.

LINES 285-294:

In reporting your final homogenization temperature, you also have to mention the mode of homogenization: to liquid or to vapour.

Fig 7a:

Although, I can see why you concluded your inclusions are primary, a better zoomed-in photograph is required to demonstrate your position. 

LINE 310: The sentence is not needed as it is repeated on the following lines.

LINES 339-340: 

This is interpreting the result and should be in discussion.

"CO2 has negative effect on quartz solubility regardless of pressure and temperature (Figure 10a - c), due to its non-electrolyte nature that reduces silica hydration degree.”

LINES 351-352: Rephrase.

LINES 358-359: This line is interpretation and should be in discussion.

"because of the expanding of fluid molar volumes.”

LINES 408-409:

It’s unclear what the consistent conclusions are? 

Also this is inconsistent with lines 45-47, where stated "By contrast, quartz solubility analysis have been much less employed in the studies of orogenic intrusion-related gold deposits, leading to less knowledge of the veining processes that are responsible for gold deposition”.

LINES 414-416: So in this case both CL and vein technology have consistent interpretations? I don’t seem to understand the point you are making here. 

LINES 431-432: Need better description/images of the more “intimate and reliable spatial relations between quartz generations with metal sulfides and gold

LINE 433: Which quartz?  "The dark CL of these quartz” 

LINES 434: How did you interpret “lower T” on darker CL?

Lines 440-442: You provided no evidence that the boiling fluid inclusion assemblage is of “primary’ origin, and therefore cannot conclude that Q2b was precipitated from such fluids.

Round 2

Reviewer 1 Report

The manuscript 'Auriferous quartz veining due to decompressional cooling and CO2 content variations revealed by quartz solubility, SEM-CL and fluid inclusion analyses (the Linglong goldfield) ' has much improved and is ready for publication.